# Learning Versatile Skills with Curriculum Masking

**Yao Tang**[*1]    **Zhihui Xie**[*1]    **Zichuan Lin**[2]    **Deheng Ye**[2]    **Shuai Li**[#1]

[1]Shanghai Jiao Tong University    [2]Tencent AI Lab

## Abstract

Masked prediction has emerged as a promising pretraining paradigm in offline reinforcement learning (RL) due to its versatile masking schemes, enabling flexible inference across various downstream tasks with a unified model. Despite the versatility of masked prediction, it remains unclear how to balance the learning of skills at different levels of complexity. To address this, we propose **CurrMask**, a curriculum masking pretraining paradigm for sequential decision making. Motivated by how humans learn by organizing knowledge in a curriculum, CurrMask adjusts its masking scheme during pretraining for learning versatile skills. Through extensive experiments, we show that CurrMask exhibits superior zero-shot performance on skill prompting tasks, goal-conditioned planning tasks, and competitive finetuning performance on offline RL tasks. Additionally, our analysis of training dynamics reveals that CurrMask gradually acquires skills of varying complexity by dynamically adjusting its masking scheme. Code is available at here.

## 1 Introduction

Humans distinguish themselves from machines by their capacity to adapt and generalize. One crucial factor behind this discrepancy is the drive to acquire reusable knowledge (e.g., concepts and behaviors) even in the absence of explicit reward (White, 1959). This has motivated research in unsupervised reinforcement learning (RL) (Laskin et al., 2021; Chebotar et al., 2021), in which the agent is required to learn from reward-free offline data (Carroll et al., 2022; Schwarzer et al., 2021) or online interaction (Liu and Abbeel, 2021; Yarats et al., 2021) for pretraining.

To build generic decision-making agents, great efforts have been made recently to apply self-supervised learning objectives for unsupervised offline pretraining (Schwarzer et al., 2021; Sun et al., 2023). Among these studies, one popular approach is *masked prediction*, a simple but versatile self-supervision framework that has proven its effectiveness in domains like language (Devlin et al., 2019) and vision (He et al., 2022). By masking a portion of the input trajectory and predicting it conditioned on the remaining unmasked tokens, the model can not only capture rich representations but also learn transferable behaviors. For example, given a masked trajectory $(s_1, \texttt{[MASK]}, s_2, a_2, \texttt{[MASK]}, a_3)$, a model learned by masked prediction is forced to reason about both dynamics (i.e., masked state $s_3$) and behaviors (i.e., masked action $a_1$).

Given the effectiveness of masked prediction, an important question arises: how can we *design* and *arrange* the masking schemes for decision-making data to maximize its benefits? To explore this question, our research stems from the finding that models trained with token-wise random masking, a widely adopted masking strategy in natural language modeling, fall short in modeling long-term dependencies (see Figure 4). While randomly masked words describe semantically dense information, state-action sequences in decision-making data contain heavy information redundancy (e.g., consecutive states are usually similar). This causes the model to predict masked tokens solely based on their neighboring unmasked tokens. Besides, state-action sequences naturally come with a

---

[*]Equal contribution.

[#]Correspondence to: Shuai Li⟨shuaili8@sjtu.edu.cn⟩.

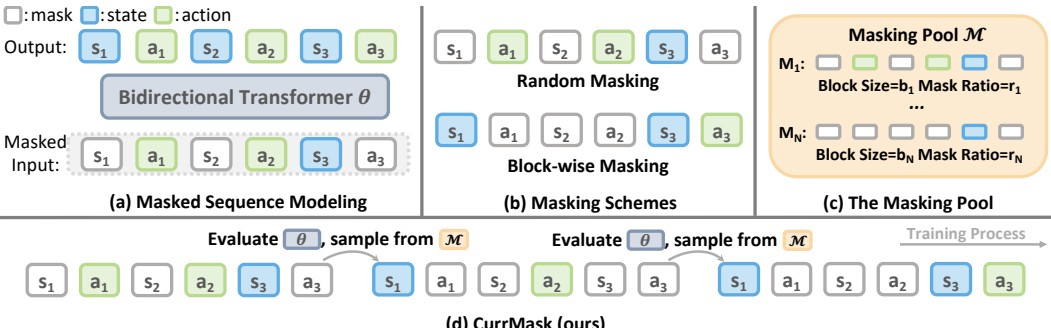

Figure 1: **Illustration of CurrMask**. Based on the framework of masked prediction modeling in (a), the block-wise masking scheme in (b), we design a masking pool $\mathcal{M}$ in (c) consisting of masking schemes at different levels of complexity, which CurrMask evaluates the learning progress of the model $\theta$ and samples masking schemes from during pretraining in (d).

unique pattern of interleaved modality, unlike single-modal word sequences. These discrepancies in *information density* and *sequence pattern* require a reevaluation of masking scheme design for RL pretraining.

To this end, we propose CurrMask to address the above considerations. CurrMask leverages block-wise masking and a curriculum-based approach arranging the order of different masking schemes to boost pretraining. An illustration of our approach is shown in Figure 1. First, we posit that masking for decision-making data needs to be designed in blocks rather than tokens. A block of consecutive state and action tokens forms a semantic entity of *skill* (Ajay et al., 2021; Pertsch et al., 2021). For example, applying block-wise masking of size 3 results in $(s_1, \texttt{[MASK]}, \texttt{[MASK]}, \texttt{[MASK]}, s_3, a_3)$. By using a block-wise masking scheme, the model is compelled to prioritize global dependencies over basic local correlations when making masked predictions. Moreover, the combination of multiple block sizes and mask ratios incentivizes the model to acquire adaptable skills that can be utilized for diverse downstream tasks.

With different block-wise masking schemes representing skills at varying levels of complexity, we further propose a curriculum-based approach to arrange the order of learning them for better capturing semantically meaningful relations. Our main intuition is that since humans obtain knowledge by organizing it into different subjects and learning the entire curriculum in an easy-to-hard order, the ability of long-horizon reasoning can be developed by first learning how to act locally. This motivates us to measure the learning progress and schedule proper masking schemes at regular intervals to boost pretraining. Ideally, the curriculum can boost the pretrained model performance on downstream tasks. However, obtaining downstream information is usually intractable during pretraining. Instead, we consider the pretraining task, i.e., the reconstruction loss, as a proxy reference for learning progress. Utilizing this proxy for skill learning, the goal of selecting masking schemes can be interpreted as maximizing the decrease in reconstruction loss(Graves et al., 2017). Considering the exploitation of the reconstruction loss decrease and the uncertainties in pretraining that require exploration of different masking schemes, the masking scheme selection task can be formulated as a multi-armed bandit problem (Lattimore and Szepesvári, 2020), where each arm represents a masking scheme.

We aim to enable the model to acquire versatile skills through CurrMask, meaning that the pretrained model becomes adaptable to different potential downstream tasks. We conduct a series of empirical studies on various MuJoCo-based control tasks, including locomotion and robotic arm manipulation. Our results demonstrate that CurrMask enables the learning of a versatile model that achieves superior performance in zero-shot skill prompting, zero-shot goal-conditioned planning as well as competitive fine-tuning performance on offline RL. Further analysis reveals that CurrMask effectively mitigates the issue of local correlations and is better at capturing long-term dependencies. These findings shed light on the design and arrangement of masking schemes that can effectively balance the acquisition of reusable skills at varying levels of temporal granularity and complexity.

## 2   Related Work

**Masked Prediction as a Self-Supervision Task.** Masked prediction requires the model to predict a missing portion of the input that has been held out. Pretraining via masked prediction has been

explored in natural language processing (Devlin et al., 2019; Joshi et al., 2020), computer vision (He et al., 2022; Bao et al., 2022; Xie et al., 2022), and decision making (Liu et al., 2022; Carroll et al., 2022; Sun et al., 2023; Wu et al., 2023; Boige et al., 2023). One of the most notable applications is masked language modeling (Devlin et al., 2019) for learning transferable text representations. Recently, it has been shown that masked prediction can also facilitate decision making, either by training visual backbones (Radosavovic et al., 2023; Seo et al., 2023) or by learning temporal information (Liu et al., 2022; Carroll et al., 2022; Sun et al., 2023; Wu et al., 2023; Boige et al., 2023). In line with this research direction, we investigate the impact of masking schemes on pretraining.

**Masking Schemes.** For masked prediction, a central question is *what is masked*. One common scheme is to randomly mask some of the tokens from the input. Apart from random masking, recent studies have proposed attention-guided masks (Li et al., 2021; Kakogeorgiou et al., 2022; Li et al., 2022) and adversarial masks (Shi et al., 2022; Tomar et al., 2023) to force the model to focus specific parts of the input for better performance. In the domain of decision making, previous work usually considers the simplest random masking scheme (Liu et al., 2022), manually designed task-specific masks (Carroll et al., 2022), random autoregressive masking (Wu et al., 2023) or a combination of random masking with other representation learning objectives (Sun et al., 2023). In this work, we aim to illustrate the connection between masking schemes and skill learning, in search for a proper automatic learning curricula for masked prediction for decision-making data.

**Unsupervised RL Pretraining.** Our work also falls into the category of extracting prior knowledge without extrinsic human supervision for sample-efficient RL. Previous work has vastly studied reward-free RL, in which the agent can interact with the environment in the absence of rewards (Eysenbach et al., 2019; Yarats et al., 2021). Another setting is to utilize unlabeled offline data for representation learning (Schwarzer et al., 2021; Stooke et al., 2021) or skill learning (Ajay et al., 2021; Jiang et al., 2022; Xie et al., 2023). Masked prediction presents a promising framework to enjoy the best of both world. Our work focuses on leveraging masked prediction for unsupervised RL pretraining.

**Curriculum Learning.** Inspired by how humans learn faster when knowledge is ordered by easiness, curriculum learning (Elman, 1993; Bengio et al., 2009) has been formulated for machine learning algorithms to improve training efficiency. While curriculum learning has been actively explored in the context of online RL (Jabri et al., 2019; Fang et al., 2021), in this work we show that offline RL pretraining also benefits from a proper learning curriculum.

## 3 Preliminaries

To better contextualize our method, we provide an overview of essential background knowledge on masked prediction and curriculum learning in this section.

### 3.1 Masked Prediction

Let $\tau = (s_t, a_t)_{t=1}^T = (s_1, a_1, s_2, a_2, \cdots, s_T, a_T)$ denote a trajectory consisting of state-action sequences and $\mathcal{D}$ denote the training dataset. The self-supervised task of masked prediction is to reconstruct $\tau$ from a masked view $\mathtt{masked}(\tau)$, where $\mathtt{masked}(\cdot)$ represents a specific masking function. For example, if $\mathtt{masked}(\cdot)$ represents a deterministic scheme that masks the initial and final actions of the input, the resulting masked trajectory is $\mathtt{masked}(\tau) = (s_1, [\mathtt{MASK}], s_2, a_2, \cdots, s_T, [\mathtt{MASK}])$. Here, $[\mathtt{MASK}]$ represents a special learnable token. The learning objective is then given by:

$$\max_{\theta} \mathbb{E}_{\tau \sim \mathcal{D}} \sum_{t=1}^{T} \log P_{\theta}\left(s_t, a_t \mid \mathtt{masked}(\tau)\right),$$

where $P_\theta$ is parameterized by a bidirectional transformer (Devlin et al., 2019). By reconstructing state-action sequences, the model learns to reason over temporal dependencies.

Importantly, the choice of $\mathtt{masked}(\cdot)$ specifies a concrete task the model is trained on. Therefore, it is crucial to design an appropriate masking scheme that enables learning of general relationships in state-action sequences. This goal boils down to two aspects: 1) *how much is masked*, and 2) *what is masked*. For the former, it has been shown that a high mask ratio (e.g., 95%) is meaningful for decision-making data due to its low information density (Liu et al., 2022). For the latter, since it is undesirable to specify the tasks of interest when pretraining, the random masking scheme is widely

used (i.e., uniformly sampling a subset of tokens to mask). These principles form the basis of our proposed masking approach, which is elaborated in Section 4.

## 3.2 Automated Curriculum Learning

Automated curriculum learning considers how to arrange the order of tasks during training by adapting the selection of learning scenarios to match the learner's abilities. Consider a series of tasks represented by loss functions $\mathcal{L}_1, \ldots, \mathcal{L}_K$. The objective is to find a time-varying sequence of tasks to accelerate training. To this end, a proper automatic curriculum needs to specify two factors: 1) how to measure *learning progress*, in order to adjust its task schedule dynamically, and 2) how to perform *task selection* based on progress signals. We describe our design in Section 4.

# 4 Curriculum Masked Prediction

In this section, we describe the proposed approach, CurrMask, for unsupervised RL pretraining. Algorithm 1 summarizes the overall pipeline. At the core of CurrMask is masked prediction as a versatile self-supervised learning objective and an automatic learning curriculum over masking schemes to enable fast skill discovery. Once pretrained on offline data, CurrMask can perform various downstream tasks in a zero-shot manner, or be finetuned for policy learning. In the following, we elaborate the design of CurrMask and provide sufficient explanation.

## 4.1 Block-wise Masking Enhances Long-term Reasoning

Our research is based on the discovery that models trained using random masking, a commonly used strategy in natural language modeling, fall short in capturing long-term dependencies (see Figure 4). This is undesirable for decision-making agents that maximize long-term reward. To overcome this issue, CurrMask applies the block-wise masking scheme (Joshi et al., 2020; Bao et al., 2022) that masks the trajectory in blocks instead of individual tokens. By doing so, CurrMask pushes the model to focus on semantically meaningful abstractions rather than simple local correlations. Predicting missing blocks of state-action sequences also resembles multi-step inverse dynamics models (Lamb et al., 2022), which has been shown to learn robust representations for decision making. We present pseudocode of our block-wise masking implementation in Appendix A.

Blocks consisting of consecutive states and actions also form a notion of *skills* or *primitives*. Prior works in offline skill discovery (Ajay et al., 2021; Jiang et al., 2022) typically use variational inference to partition trajectories into skills. In this work, we argue that masked prediction with block-wise masking represents an alternative approach for offline skill discovery. The link between masked prediction and skill discovery inspires us to explore automatic curricula that can aid in learning skills.

## 4.2 Learning over a Mixture of Masking Schemes

The block size explicitly determines the level of temporal granularity for masked prediction. To capture both short-term and long-term temporal dependencies, CurrMask employs a combination of masking schemes with varying block sizes and mask ratios during pretraining.

Given a set of masking schemes $\mathcal{M}$ where $|\mathcal{M}| = K$, we define the loss function for masked prediction task $k$ as:

$$\mathcal{L}_k(\tau; \theta) = \sum_{t=1}^{T} \log P_\theta \left( s_t, a_t \mid \texttt{masked}_k(\tau) \right),$$

where $\texttt{masked}_k \in \mathcal{M}$ denotes a specific masking scheme. CurrMask aims to minimize the multi-task learning objective $\mathcal{L}_{\text{target}}(\tau; \theta) = \frac{1}{K} \sum_{k=1}^{K} \mathcal{L}_k(\tau; \theta)$.

## 4.3 Automated Curriculum Learning Boosts Training Efficiency

A key feature of mixed masking schemes is their inherent variability in complexity. Intuitively, the ability of reasoning over global dependencies can be developed by first learning how to plan within a short horizon. This motivates us to consider curriculum learning to facilitate masked prediction.

---

**Algorithm 1:** Curriculum Masking

---
**Input :** masking pool $\mathcal{M}$ with cardinality $K$, uniformly initialized probability distribution
$\pi_\mathbf{w}$, training steps $T$, evaluation interval $I$, offline training dataset $\mathcal{D}$, validation
dataset $\mathcal{D}_{val}$, bidirectional transformer $P_\theta$

Initialize exponential weights $w_k \leftarrow 1$ for $k = 1, \ldots, K$;
Compute initial target loss $\mathcal{L}_{\text{target}}(\theta)$ on validation dataset $\mathcal{D}_{val}$ with Equation 4.2;
Sample initial masking scheme $k \sim \pi_\mathbf{w}$;
**for** $t \leftarrow 1 \ldots T$ **do**
    /* Training                                                   */
    Sample masks with $k$, compute training loss $\mathcal{L}_k$ on $\tau \sim \mathcal{D}$;
    Update $\theta$ by gradient descent;
    **if** $t \mod I = 0$ **then**
        /* Evaluation & update of masking scheme                  */
        Compute target loss $\mathcal{L}_{\text{target}}(\theta)$ on $\mathcal{D}_{val}$;
        Calculate reward $r$ using scaled target loss decrease in Equation 1;
        Update weights $\mathbf{w}$ with Equation 4;
        Update probability distribution $\pi_\mathbf{w}$ with Equation 3;
        Update masking scheme $k \sim \pi_\mathbf{w}$;
    **end**
**end**

---

By scheduling masking schemes in a meaningful order, we expect that the model will learn more efficiently and quickly during training.

**Evaluation of Learning Progress.** The first factor to determine is the measure of learning progress. Ideally, we would like the curriculum to maximize the rate at which the model learns to solve downstream tasks. However, it is usually intractable to measure without downstream task information. Hence, we consider *target loss decrease* (Graves et al., 2017) as a proxy signal for learning progress:

$$r = f_{\text{scale}}(\mathcal{L}_{\text{target}}(\theta) - \mathcal{L}_{\text{target}}(\theta')), \tag{1}$$

where $\theta$ and $\theta'$ denote the model parameters before and after training on a masking scheme for an interval $I$, respectively. The underlying rationale is that the most efficacious masking scheme is evidenced by the greatest target loss decrease observed before and after training on it. To alleviate the issue of time-varying magnitudes, we follow Graves et al. (2017) to rescale values into $[-1, 1]$ using the 20-th percentile $r_t^{\text{lo}}$ and 80-th percentile $r_t^{\text{hi}}$ of unscaled history values $R_t = \{\hat{r}_i | i = 0, I, 2I, ..., t\}$, where $I$ represents the evaluation interval and $t$ is some interval timestep (i.e., $t$ is divisible by $I$):

$$f_{\text{scale}}(\hat{r}_t) = \max(-1, \min(1, \frac{2\left(\hat{r}_t - r_t^{\text{lo}}\right)}{r_t^{\text{hi}} - r_t^{\text{lo}}} - 1)). \tag{2}$$

**Task Selection.** To schedule tasks in a meaningful order, we aim to minimize $\mathcal{L}_{\text{target}}$ achieved after training on them sequentially, which is equivalent to maximizing the reward defined in Equation 1. In the selection of masking schemes, we need to balance the maximization of the reward with the uncertainties inherent in pretraining dynamics, requiring deliberate exploration among diverse masking schemes. This can be formulated a multi-armed bandit problem (Lattimore and Szepesvári, 2020), where each arm represents a masking scheme and the goal is to maximize the total reward earned over time. Since the reward distribution induced by Equation 1 shifts as the network learns, we use the EXP3 algorithm (Auer et al., 2002), a non-stochastic multi-armed bandit algorithm that mixes the probability distribution computed using exponential weights $\mathbf{w}$ with the uniform distribution constituting an $\epsilon$ fraction of the total probability distribution. The sampling probability distribution $\pi_\mathbf{w}$ for each masking scheme is then given by:

$$\pi_\mathbf{w}(i) = (1 - \epsilon)\frac{w_i}{\sum_{j=1}^{K} w_j} + \frac{\epsilon}{K} \quad i = 1, \ldots, K, \tag{3}$$

where $\omega_i$ represents the weight of the i-th arm (or masking scheme), $\omega_i'$ represents the updated $\omega_i$ in Equation 4, $K$ denotes the number of arms available in the masking scheme pool and $\pi$ represents the

sampling probability distribution for each masking scheme. This equation shows how to sample each arm (i.e., each masking scheme) using weights. The weights $\omega$ determine the probability distribution $\pi$ from which we sample the masking schemes. This probabilistic approach allows us to balance exploration and exploitation by dynamically adjusting the focus on different masking schemes based on their performance.

Each time EXP3 samples an arm $k$ by the policy $\pi_{\mathbf{w}}$, observes reward $r$, and uses the importance-weighted estimator $\hat{x}_i = \frac{\mathbb{I}\{i=k\}r}{\pi_{\mathbf{w}}(i)}$ to update its weights according to the following formula:

$$w'_i = w_i \exp\left(\gamma \hat{x}_i / K\right) \quad i = 1, \ldots, K. \tag{4}$$

This equation describes how we update the arm weights based on the observed rewards, which in this context are the computed pretraining progress. By updating the weights $\omega$ according to the rewards, we ensure that the more effective masking schemes (those that lead to better pretraining progress) are sampled more frequently in subsequent iterations. This adaptive mechanism helps the model to learn an effective masking curriculum over time. As such, exponential growth significantly increases the probability of choosing good arms (i.e., masking schemes). Please see Appendix C for discussions on how CurrMask addresses non-stationarity.

# 5 Experiments

In this section we conduct an empirical study to answer the following questions: **(Q1)** Can CurrMask learn a versatile model that achieves good performance on a variety of downstream tasks, both in zero-shot and finetuning scenarios? **(Q2)** What role do block-wise masking and masking curricula play in CurrMask? **(Q3)** Does CurrMask better capture long-term temporal dependencies, and if so, what mechanism within CurrMask facilitate this capability?

## 5.1 Environment Setup

We evaluate our method on a set of environments from the DeepMind control suite (Tunyasuvunakool et al., 2020). Each environment has several tasks specified by how the reward function is defined. Specifically, we consider a total of 9 tasks that are associated with 3 different environments (`walker`, `quadruped` and `jaco`). At evaluation, we test how well the model pretrained for each environment on offline datasets adapts to different downstream tasks. For more details and experimental results, please refer to Appendix B.2 and Appendix D, respectively.

**Environments.** The `walker` environment consists of 3 locomotion tasks (`run`, `stand`, and `walk`) and the `quadruped` environment provides 2 locomotion tasks (`run` and `walk`). All the tasks provide a dense reward measure of task completion. For example, task `run` provides rewards encouraging forward velocity. We also conduct experiments on `jaco`, which is an environment for robot arm manipulation including 4 reaching tasks (`bottom_left`, `bottom_right`, `top_left`, and `top_right`). These tasks are sparse-reward tasks given that nonzero rewards are provided only when the current position is within a certain distance threshold of the target position.

**Dataset Collection.** For each environment, we construct a multi-task dataset by collecting trajectories of 12M steps from the replay buffer of TD3 agents (Fujimoto et al., 2018). This collection procedure ensures that the pretraining dataset contains experiences of varying quality. For zero-shot evaluation, we additionally construct a validation set for each environment using the same protocol but with different random seeds, following the setting in the prior work (Liu et al., 2022).

**Implementation Details.** We consider multiple mask ratios $\mathcal{R} = \{15\%, 35\%, 55\%, 75\%, 95\%\}$ and multiple block sizes $\mathcal{B} = \{1, 2, \ldots, 20\}$ to construct the masking pool $\mathcal{M} = \{(r, b) \mid r \in \mathcal{R}, b \in \mathcal{B}\}$ ($|\mathcal{M}| = 100$) for CurrMask. For all the evaluated masked prediction methods, we use the same bidirectional encoder-decoder transformer architecture with a 3-layer encoder and a 2-layer decoder, following prior works (He et al., 2022; Liu et al., 2022). The encoder input is unmasked states and actions and the decoder input is the whole trajectory including both masked and unmasked tokens. The evaluated autoregressive baselines consist of 5 layers for fair comparison.

**Baselines.** We compare CurrMask with the following baselines: **MaskDP** (Liu et al., 2022) samples a mask ratio from $\mathcal{R}$ and randomly masks a portion of individual tokens in each training step; **MTM** (Wu et al., 2023) adopts random autoregressive masking which first randomly samples masked tokens and all future tokens of the last masked token are masked; **Mixed** randomly samples a masking scheme from $\mathcal{M}$ with a uniform distribution; **Mixed-prog** uses a manually designed mask curriculum that progressively increases the block size during pretraining, divided into four stages. In each

Table 1: **Skill prompting results.** We report the zero-shot performance of models pretrained with different masking schemes. Results are averaged over 10 random seeds. The best and second results are bold and underlined, respectively.

| Reward ↑ | walker_s | walker_w | walker_r | quad_w | quad_r | jaco_bl | jaco_br | jaco_tl | jaco_tr | Average |
|---|---|---|---|---|---|---|---|---|---|---|
| MaskDP | 103.2 ±2.6 | 58.4 ±2.3 | 29.3 ±1.4 | 36.6 ±2.2 | 45.1 ±2.4 | 58.1 ±4.4 | 58.4 ±3.0 | 56.9 ±3.9 | 64.0 ±3.3 | 56.7 |
| MTM | 107.1 ±2.8 | 58.8 ±2.7 | 27.3 ±1.4 | 37.1 ±2.3 | 42.8 ±2.7 | 72.0 ±4.5 | 71.8 ±3.9 | 72.5 ±5.1 | 77.6 ±3.5 | 63.0 |
| Mixed-inv | 103.3 ±3.1 | 59.5 ±3.0 | 23.8 ±1.2 | **45.1** ±3.0 | 43.2 ±2.9 | 51.8 ±3.3 | 53.0 ±2.8 | 56.8 ±3.6 | 59.7 ±4.8 | 55.1 |
| Mixed-prog | 103.5 ±2.4 | 55.0 ±2.8 | 25.8 ±1.2 | 40.5 ±1.8 | 45.6 ±2.2 | 85.3 ±5.5 | 85.4 ±3.7 | 84.2 ±4.8 | 88.5 ±3.7 | 68.2 |
| Mixed | 110.8 ±2.2 | 54.2 ±2.0 | 30.5 ±1.2 | 43.3 ±2.7 | 51.3 ±2.8 | 66.0 ±6.4 | 61.6 ±3.7 | 62.3 ±3.6 | 66.5 ±4.0 | 60.7 |
| GPT | 101.8 ±2.9 | 34.6 ±1.3 | 21.6 ±1.0 | 41.9 ±2.9 | 48.8 ±3.2 | 86.1 ±5.7 | 83.1 ±2.7 | 83.9 ±5.1 | 85.7 ±3.0 | 65.3 |
| CurrMask | **111.2** ±2.4 | **79.9** ±1.2 | **38.9** ±1.9 | 38.0 ±2.2 | 51.0 ±3.4 | **88.4** ±5.1 | **88.5** ±3.6 | **86.0** ±4.3 | **92.9** ±3.5 | 75.0 |

stage, block sizes are sampled from $\{1, 2, \ldots, 5\}$, $\{1, 2, \ldots, 10\}$, $\{1, 2, \ldots, 15\}$, and $\{1, 2, \ldots, 20\}$ respectively, while the mask ratio is sampled from $\mathcal{R}$; **Mixed-inv** adopts an inverted approach to the mask curriculum of `Mixed-prog`, utilizing the block size pools in reverse order. Apart from masked prediction baselines, we also compare with **GPT** and **Goal-GPT**, which are auto-regressive models similar to GPT (Brown et al., 2020). `GPT` takes the past states and actions as inputs and outputs the next state or action. To accommodate downstream tasks that require information about the goal state, `Goal-GPT` is modified to take a goal state as well as past states and actions as input and predict the next state or action to reach the goal.

## 5.2  Downstream Tasks

To demonstrate the versatility of CurrMask, we consider various downstream tasks that require different capabilities, including zero-shot inference by specifying certain masking schemes (i.e., skill prompting and goal-conditioned planning) and adaptation via finetuning (i.e., offline RL).

**Skill Prompting.** A unified model trained on diverse multi-task data is expected to acquire various skills that can be invoked to perform certain tasks. Skill prompting tests this ability by requiring the model to generate consecutive behaviors given a short state-action sequence. An input example of which prompt context length is 3 looks like $(s_0, a_0, s_1, a_1, s_2, a_2, [\texttt{MASK}], [\texttt{MASK}], ..., [\texttt{MASK}])$ and during evaluation, we set the environment state to $s_2$, perform predicted actions and record rewards. For each task, we sample prompt contexts of length eight from the validation set, and evaluate the quality of the generated trajectory of length 120 by its task rewards.

**Goal-conditioned Planning.** Another type of downstream task we consider is goal-conditioned planning. Starting from a given state, the model needs to roll out actions that can achieve target goals within a number of steps. We condition the pretrained model a start state and four goal states to evaluate the model's capability to generate long-term plans. With the input pattern of $(s_{start}, \texttt{MASK}, ..., \texttt{MASK}, s_{goal1}, \texttt{MASK}, ..., \texttt{MASK}, s_{goal2}, \texttt{MASK}, ...)$, we set the environment state to $s_{start}$ perform the predicted actions. The performance is assessed by the L2 distance between each goal and the state that is achieved by executing the predicted actions within the given time budget and is closest to the goal. We choose the goal states at distances of 20, 40, 60, and 80 future timesteps from the start states to evaluate the long-term planning capability of model.

**Offline RL.** Finally, we study if the representations learned by CurrMask can accelerate offline RL. For each task, we add a critic head and actor head on top of the encoder and run TD3 (Fujimoto et al., 2018) to perform offline RL training, following prior work (Liu et al., 2022). Although TD3 is originally designed as an off-policy RL algorithm, Yarats et al. (2022) show that it achieves very competitive performance on offline datasets of diverse behaviors. The offline dataset is collected from the entire replay buffer of a ProtoRL agent (Yarats et al., 2021) trained for 2M environment steps. Notably, the datasets consist of highly exploratory data, which emphasizes the importance of having good representations.

## 5.3  Main Results

We test the versatility of CurrMask over a variety of downstream tasks, in answer to **Q1** and **Q2**.

Table 2: **Goal-conditioned planning results.** We report the zero-shot performance of models pretrained with different masking schemes. Results are averaged over 20 random seeds. The best and second results are bold and underlined, respectively. The lower the better.

| Distance ↓ | walker_s | walker_w | walker_r | quad_w | quad_r | jaco_bl | jaco_br | jaco_tl | jaco_tr | Average |
|---|---|---|---|---|---|---|---|---|---|---|
| MaskDP | 4.85 ±0.48 | 10.10 ±0.27 | 15.52 ±0.39 | 20.71 ±0.69 | 21.62 ±0.79 | 1.42 ±0.05 | 1.42 ±0.05 | 1.39 ±0.06 | 1.40 ±0.06 | 8.71 |
| MTM | 6.05 ±0.61 | 12.20 ±0.41 | 17.92 ±0.55 | 23.93 ±0.70 | 25.09 ±0.80 | 2.38 ±0.08 | 2.42 ±0.10 | 2.35 ±0.07 | 2.30 ±0.10 | 10.59 |
| Mixed-inv | 5.32 ±0.53 | 11.25 ±0.31 | 16.51 ±0.51 | 22.63 ±0.74 | 23.31 ±0.77 | 1.55 ±0.06 | 1.53 ±0.05 | 1.57 ±0.07 | 1.53 ±0.08 | 9.47 |
| Mixed-prog | 4.96 ±0.48 | 10.18 ±0.28 | 15.77 ±0.48 | 23.49 ±0.72 | 24.28 ±0.86 | 1.46 ±0.04 | 1.44 ±0.04 | 1.44 ±0.05 | 1.44 ±0.09 | 9.38 |
| Mixed | 4.83 ±0.47 | 10.15 ±0.28 | 15.47 ±0.46 | 20.67 ±0.73 | 21.66 ±0.75 | 1.47 ±0.06 | 1.47 ±0.04 | 1.43 ±0.06 | 1.44 ±0.08 | 8.73 |
| Goal-GPT | 7.47 ±0.74 | 15.15 ±0.41 | 21.04 ±0.60 | 27.36 ±0.77 | 28.76 ±0.90 | 3.34 ±0.10 | 3.58 ±0.11 | 3.26 ±0.15 | 3.50 ±0.11 | 12.61 |
| CurrMask | 4.85 ±0.47 | **9.90** ±0.27 | 15.31 ±0.49 | **20.47** ±0.71 | **21.39** ±0.67 | **1.39** ±0.05 | **1.38** ±0.04 | **1.33** ±0.05 | **1.34** ±0.07 | **8.60** |

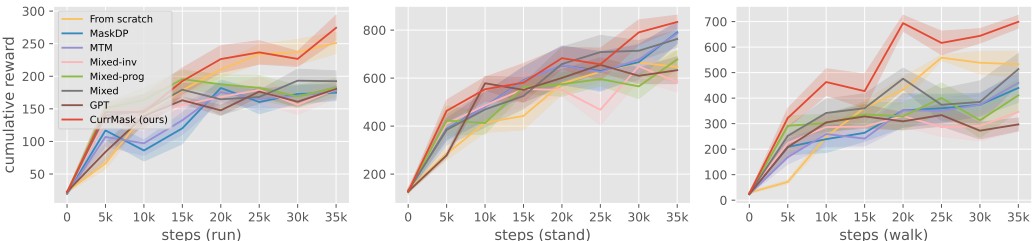

Figure 2: **Offline RL results.** We report the finetuning performance of models pretrained with different masking schemes. Results are averaged over 10 random seeds.

**Skill Prompting Results.** Table 1 summarizes the zero-shot performance for skill prompting. We observe that CurrMask achieves the best performance in 7 out of 9 tasks and also attains the highest average performance, outperforming `MaskDP` by a margin of 32%. suggesting that CurrMask is proficient in mirroring skill prompts to accomplish specific tasks. Besides, other baseline methods that incorporate block-wise masking (i.e., `Mixed`, `Mixed-prog`, and `Mixed-inv`) generally outperform random masking. This matches our expectation that blocks form more semantically meaningful entities than individual tokens and can be utilized by masked prediction to facilitate skill learning. The only exception is `Mixed-inv`. The poor performance of `Mixed-inv` sends a strong signal that a proper curriculum is important for masked prediction training.

**Goal-conditioned Planning Results.** Next, we evaluate how capable CurrMask is for long-horizon planning. As shown in Table 2, CurrMask can roll out better goal-reaching trajectories than the baselines in 8 out of 9 tasks. We can observe that `Goal-GPT` exhibits the worst performance in all the tasks, suggesting that the autoregressive model falls short in downstream tasks that necessitate the simultaneous use of bidirectional information, compared to directional models. Another notable observation is that, in contrast to skill prompting results, `Mixed, Mixed-prog` and `Mixed-inv` have worse performance than `MaskDP`. This indicates that the superior performance of CurrMask is not only due to block-wise masking but rather a consequence of dynamically balancing what to mask during training.

**Offline RL Results.** Finally, we present offline RL results in Figure 2. Compared with learning from scratch, learning with pretrained representations obtained by CurrMask results in training speedup and performance improvement. We observe that CurrMask generally outperforms other masked pretraining baselines as well as autoregressive architecture based baselines, which suggests that CurrMask not only learns diverse skills but also extracts transferable representations for policy learning. It should also be noted that in some cases (e.g., `walk` and `run`) pretraining with `GPT` leads to diminished performance for finetuning, whereas CurrMask is generally more stable.

### 5.4 Analysis

In this section, we investigate several aspects of CurrMask to further answer **Q2** and **Q3**.

**Impact of Block-wise Masking.** To better understand how block-wise masking contributes to CurrMask, we conduct an ablation study on the choice of block sizes. Figure 3a shows the influence of the block size, where masked prediction is combined with a fixed block size and mask ratios randomly sampled from $\mathcal{R}$. With block-wise masking, masked prediction benefits from larger block

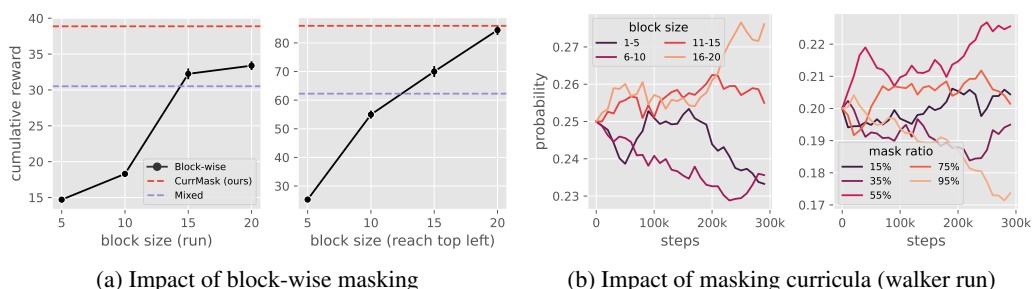

(a) Impact of block-wise masking      (b) Impact of masking curricula (walker run)

Figure 3: **Both block-wise masking and curriculum masking contribute to CurrMask's performance.** Left: the performance of zero-shot skill prompting as a function of fixed block size. Right: the probabilities of choosing different block sizes and mask ratios during pretraining with CurrMask.

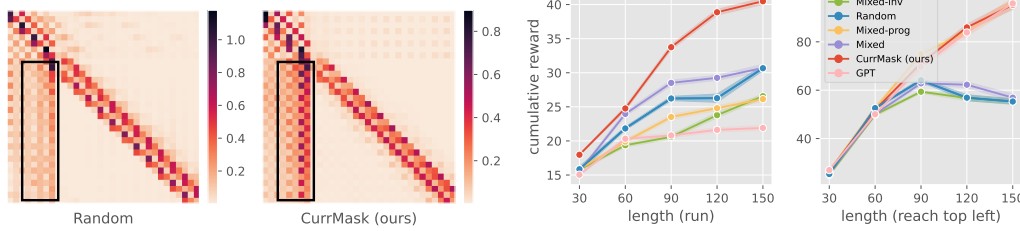

(a) Attention maps (reach top left)  (b) Skill prompting performance vs. rollout length

Figure 4: **Analysis of long-term prediction capability.** Left: We visualize the attention map (L2-normalized over different heads) of the first decoder layer, when the model is conducting skill prompting. The horizontal axis represents the keys, and the vertical axis represents the queries. Right: the performance of zero-shot skill prompting as a function of rollout length (tasks: `walker run` and `jaco reach top left`).

sizes to perform zero-shot skill prompting. Besides, mixing different block sizes uniformly for training, referred to as `Mixed`, leads to average performance. By leveraging a proper combination of masking schemes with different block sizes, CurrMask boost the skill prompting performance matching or exceeding the best results among all the block-wise masking schemes shown in Figure 3a. This indicates that the block size does have a great impact over final performance, and a proper learning curriculum can be also crucial.

**Impact of Masking Curricula.** Another important question is whether good masking curricula should be determined manually or found adaptively during training. We would like to emphasize that `Mixed-prog` does not consistently lead to performance improvements compared to `Random`. Specifically, in most goal-conditioned planning tasks and in offline RL (shown in Figure 2) and skill prompting tasks of the walker domain, Mixed-prog performs worse than `Random`. In contrast, for CurrMask, we consistently observe improvements over `Random`.

We provide additional experimental results to better illustrate that CurrMask is not just rediscovering the programmatic curriculum. Figure 5 displays the skill prompting results versus training steps during pretraining, revealing noticeable differences in skill learning progress between `Mixed-prog` and CurrMask. This comparison emphasizes that a proper masking scheme and curriculum-based pretraining progress are unlikely to be predetermined and highlights the benefits of CurrMask for its adaptivitity.

**Evaluation of Long-term Prediction.** One of the most important intuition behind CurrMask is that block-wise masking can enhance the model's capability to capture long-term dependencies. To verify this, we look into the attention maps during prediction with skill prompts even when they are far from current timesteps. Figure 4a shows how CurrMask predicts the attention map for all 32 tokens based on the first 8 unmasked tokens and the subsequent 24 masked tokens. The vertical axis represents the query, and the horizontal axis represents the key. The black-boxed area highlights the model's dependency on the keys of the first 8 unmasked tokens when predicting the attention map for the subsequent 24 masked tokens. We notice significant differences in how the approaches use the prompt. It can be observed that the attention of CurrMask in the bounded region is more pronounced, indicating that CurrMask is better at recalling prompt context when predicting the future compared to random masking. Besides, the predictions of CurrMask attend to prior actions more than they do to prior states. These findings support our intuition that CurrMask is more effective at extracting useful long-term dependencies. We offer a more comprehensive assessment in Appendix E.

This discrepancy in attention patterns is further validated by the performance of long-horizon skill prompting. Figure 4b shows the skill prompting performance as a function of the rollout length. We observe that CurrMask outperforms the baselines significantly when the rollout length is extended. Notably, `Random`, `Mixed` and `Mixed-inv` have degenerated performance for long rollouts on the right task, supporting our hypothesis that CurrMask acquires non-trivial long-term prediction capacity.

**Visualization of Masking Curricula.** Next, we investigate how automatic curricula steer masking schemes during training. Figure 3b visualizes the time-varying probabilities of choosing different block sizes and mask ratios during CurrMask pertaining. We can see that CurrMask gradually increases the probability of choosing large block sizes while also preferring a moderate mask ratio. The former observation reveals that CurrMask has a tendency to learn more complex skills, which aligns with our intuition. For the latter, we believe it reflects the degree of information redundancy in sequential decision-making data, also reported in previous work (Liu et al., 2022; He et al., 2022).

We also visualize the mean block size and mean mask ratio during pretraining in Appendix D.1 for further reference.

## 6  Conclusion

In this work, we propose CurrMask, a curriculum masking approach for unsupervised RL pretraining. Motivated by the unique pattern of sequential decision-making data (i.e., *low information density* and *interleaved modality*), we propose to apply block-wise masking with mixed mask ratios and block sizes to capture temporal dependencies at both short-term and long-term levels of granularity. As different masking schemes naturally vary in prediction difficulty, we consider automated curriculum learning as the inner drive to facilitate training by scheduling these schemes in a meaningful order. We show through extensive experiments that CurrMask learns a versatile model that consistently outperforms the baselines in various downstream tasks. Our analysis of the impact of block-wise masking and curriculum learning emphasizes the adaptivity of CurrMask and its superior ability to extract global dependencies.

**Limitations.** One limitation of CurrMask is the computational overhead, as CurrMask relies on an extra bandit model to schedule masking schemes for training[1]. Furthermore, the advantages offered by CurrMask could be affected by the underlying structure of the environment. This encourages us to extend our method to more challenging settings like image-based RL in future research.

## Acknowledgements

The corresponding author Shuai Li is sponsored by CCF-Tencent Open Research Fund.

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

# A Pseudocode of Block-wise Masking

Algorithm 2 demonstrates the block-wise masking mechanism, which is employed as an intermediate step for masking in CurrMask and other baseline models.

---

**Algorithm 2:** Block-wise Masking

**Input** : sequence length $L$, mask ratio $p$, block size $b$
**Output:** binary mask matrix $m \in \{0, 1\}^L$ (0 for masked, 1 for unmasked),
**if** $b = 1$ **then**
  |   **return** random_masking$(L, p)$;
**end**
$l \leftarrow p \cdot L$ ;                        /* length of masked tokens */
$c \leftarrow \lfloor \frac{L-1}{b} \rfloor$ ;               /* number of blocks in a sequence */
Initialize mask $m \leftarrow \mathbb{1}$;
Randomly choose start index $s \leftarrow$ random$(0, \cdots, b - 1)$;
Shuffle block indices $bs \leftarrow$ shuffle$(0, \ldots, c - 1)$;
Expand block indices to token indices $ts \leftarrow (i \cdot b + j + s$ for $i$ in $bs$ for $j$ in $(1, \cdots, b - 1))$;
Mask tokens $m \leftarrow$ set_zero$(m, ts[-l :])$;
**if** $len(ts) < l$ **then**
  |   Mask remaining tokens $m \leftarrow$ set_zero$(m, ((0, \cdots, L - 1) - ts)[len(ts) - l :])$;
**end**
**return** $m$;

---

# B Experimental Details

## B.1 Data Collection

**Pretraining Datasets** For all the environments, we create a multi-task dataset by gathering trajectories from the replay buffer of TD3 agents. We collect a total of 12M steps from the replay buffer for each environment. Each task in the walker environment is trained for 4M environment steps, each task in the jaco environment is trained for 3M steps and each task in the quadruped environment is trained for 6M steps. By following this procedure, we ensure that the pretraining datasets encompasses experiences of varying quality.

**Validation Datasets** For zero-shot evaluation of both skill prompting and goal-conditioned planning, we construct a separate validation set for each environment using the same collection protocol for pretraining datasets but with different random seeds.

**Training Datasets for Offline RL** Each offline dataset is obtained from the complete replay buffer of a ProtoRL agent, which was trained for 2M environment steps. For each task, the collected dataset is relabeled with task-specific rewards during offline RL. It is worth mentioning that these datasets contain highly exploratory data, emphasizing the significance of having effective representations. Table 3 summarizes the statistics.

Table 3: Episodic return statistics of training datasets used for offline RL.

| task | min | max | mean |
|------|-----|-----|------|
| stand | 27.07 | 408.59 | 198.85 |
| walk | 4.81 | 199.95 | 72.95 |
| run | 4.55 | 79.00 | 38.61 |

## B.2 Implementation Details

**Hyperparameters** Our CurrMask implementation is based on the MaskDP codebase[2]. Table 4 summarizes the hyperparameters used by CurrMask for training and evaluation.

---

[2]https://github.com/FangchenLiu/MaskDP_public

Table 4: Hyperparameters used for model training and evaluation.

| model | value |
| --- | --- |
| # encoder layers | 3 |
| # decoder layers | 2 |
| # autoregressive transformer layers | 5 |
| # attention heads | 4 |
| context length | 64 |
| hidden dimension | 256 |
| mask ratio | $[15\%, 35\%, 55\%, 75\%, 95\%]$ |
| block size | $[1, 2, \ldots, 20]$ |
| **training** | |
| optimizer | Adam |
| batch size | 384 |
| learning rate | 1e-4 |
| # gradient steps | 300k |
| EXP3 $\epsilon$ | 0.2 |
| EXP3 $\gamma$ | 0.1 |
| evaluation interval $I$ | 100 |
| # evaluation samples $N$ | 10 |
| **skill prompting** | |
| # seeds | 10 |
| # trajectories sampled per seed | 100 |
| prompt length | 8 |
| rollout length | 120 |
| **goal-conditioned planning** | |
| # seeds | 20 |
| # trajectories sampled per seed | 100 |
| **offline RL** | |
| # seeds | 10 |
| # training steps | 35k |

**Baselines and Implementation Details** Our method CurrMask focuses on masked prediction. Therefore, we aim to compare CurrMask with the most relevant skill learning methods that are also based on masked prediction. For all the baselines, we use the same model architecture and common hyperparameters as CurrMask. The implementation of `Mixed-prog` involves evenly dividing the pretraining process into four stages based on the ratio of training steps to total training steps. We sequentially apply four distinct mask schemes, denoted as $\{(r, b) \mid r \in \mathcal{R}, b \in \{1, 2, \ldots, 5\}\}, \{(r, b) \mid r \in \mathcal{R}, b \in \{1, 2, \ldots, 10\}\}, \{(r, b) \mid r \in \mathcal{R}, b \in \{1, 2, \ldots, 15\}\}$, and $\{(r, b) \mid r \in \mathcal{R}, b \in \{1, 2, \ldots, 20\}\}$. This deliberate control enables the progressive increase in block size of the mask, posing greater challenges to the training procedure as it unfolds. The implementation of `Mixed-inv` shares significant similarities with `Mixed-prog`. Both methods adopt a four-stage approach to partition the training process. The key distinction lies in the sampling of sub-sequence lengths as training progresses. In the case of `Mixed-inv`, these lengths follow a descending pattern: $\{(r, b) \mid r \in \mathcal{R}, b \in \{1, 2, \ldots, 20\}\}, \{(r, b) \mid r \in \mathcal{R}, b \in \{1, 2, \ldots, 15\}\}, \{(r, b) \mid r \in \mathcal{R}, b \in \{1, 2, \ldots, 10\}\}$, and $\{(r, b) \mid r \in \mathcal{R}, b \in \{1, 2, \ldots, 5\}\}$.
For all the evaluated bidirectional masked prediction methods, we use the same encoder-decoder transformer architecture with a 3-layer encoder and a 2-layer decoder and the same learning objective following prior works (Liu et al., 2022). The learning objective is a design choice to compute loss on the entire input (Vincent et al., 2008) or only on the masked tokens (Devlin et al., 2019). In this work, we apply the former as it has been shown to work better with sequential decision-making data (Liu et al., 2022).

**Compute Resources** CurrMask is intended to be approachable to the RL research community. The entire workflow can be executed on a single GPU. Utilizing a single RTX 3090 graphics card, the pretraining on the assembled datasets takes approximately 7-8 hours for 300k gradient steps.

**Evaluation of Skill Prompting** To facilitate skill prompting, the agent is provided with a short state-action segment randomly extracted from a trajectory in the validation dataset. The agent is

then positioned at the final state of the segment and tasked with generating subsequent behaviors in an autoregressive manner. The quality of the generated sequence is evaluated by comparing its accumulated rewards with those obtained from the rollout of an expert with advanced skills. In detail, we employ a prompt length of 8 timesteps and the initial position of each prompt is randomly sampled within the range of $[0.1 \cdot \texttt{trajectory\_length}, 0.85 \cdot \texttt{trajectory\_length}]$. Therefore, the prompt may be located at the beginning of a trajectory or skewed towards the later stages, resulting in the agent's state being in a low-speed starting phase or a high-speed running phase in the cases of the `walk/run` task.

**Evaluation of Goal-conditioned Planning**  To implement goal-conditioned planning, we randomly sample a goal context of length 100 from the trajectories in the validation set. The position of the goal is set at specific locations $[20, 40, 60, 80]$. The agent is initially placed at the starting position of the goal context, and the rollout continues for the remaining tokens. We calculate the L2 distance between each goal state and its closest state token within the rollout length as a metric for evaluation.

**Evaluation of Offline RL**  In offline RL, the main objective is to train a model to maximize the return for a specific task, as defined by a reward function. This differs from our self-supervised pretraining objective, so additional finetuning is required. To align with the RL setting, we modify the bidirectional attention mask in the transformer to a causal attention mask. This change allows the model to attend only to previous states and actions during training, simulating the sequential nature of RL tasks. We also utilize a standard actor-critic framework similar to TD3 by incorporating a critic head and an actor head on top of the pretrained encoder. The actor takes a sequence of states as input, while the critic takes a sequence of state-action pairs as input. Both components operate without any masking. Then we perform RL training using the modified architecture.

## C  Discussions on Non-stationarity

Non-stationarity is a major challenge for algorithm design in our context. We want to emphasize two important properties: 1) The reward distribution is non-stationary, and 2) Despite this, the learning process usually progresses gradually without sudden regime shifts (Zhou et al., 2021).

For the former, we want to emphasize that our method tackles the non-stationarity in two aspects. Firstly, EXP3, a special case of online mirror descent, is inherently adaptable to reward distributions that change over time. Secondly, we alleviate this issue by rescaling rewards using historical percentiles. For the latter, while abrupt distribution changes are not typically observed, we believe that our framework can easily accommodate other techniques like sliding windows and reward discounting to address significant non-stationarity.

## D  Additional Experimental Results

### D.1  Impact of Masking Curricula

In Figure 5, we plot the cumulative reward of each 30 steps of the generated trajectory. The patterns of `Mixed-prog` and CurrMask are substantially different. During the initial stage of pretraining, `Mixed-prog` (trained with small blocks only) struggles to learn skills at all levels of temporal granularity. CurrMask however exhibits faster skill acquisition and adapts its masking scheme dynamically during training.

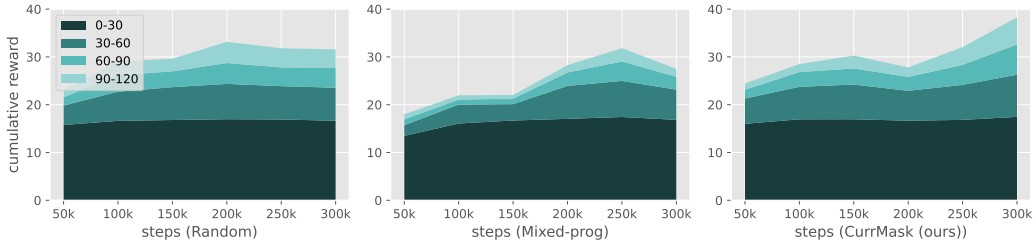

Figure 5: Skill prompting performance on Walker run in the pretraining phase.

Figure 6 illustrates the mean block size and mean mask ratio used by CurrMask during pretraining, as a function of training steps. We can see that during the pretraining process, CurrMask continually adjusts the block size and mask ratio according to the training progress. Overall, there is an upward

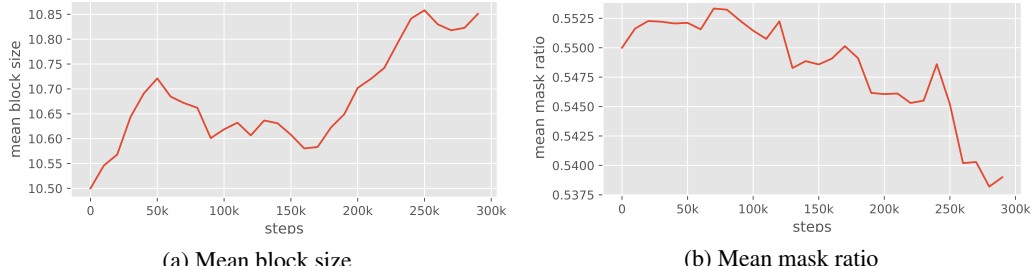

(a) Mean block size          (b) Mean mask ratio

Figure 6: Visualizations of mean block size and mean mask ratio during curriculum pretraining on walker domain.

Table 5: **Reward results.** We report the performance across Token-wise masking and Block-wise masking with different block sizes. The best and second-best results are bold and underlined, respectively.

| Reward ↑ | walker_s | walker_w | walker_r | quad_w | quad_r | jaco_bl | jaco_br | jaco_tl | jaco_tr | Average |
|---|---|---|---|---|---|---|---|---|---|---|
| Token | 103.2 ± 2.6 | 58.4 ± 2.3 | 29.3 ± 1.4 | 36.6 ± 2.2 | **45.1** ± 2.4 | 58.1 ± 4.4 | 58.4 ± 3.0 | 56.9 ± 3.9 | 64.0 ± 3.3 | 56.7 |
| Block-5 | 95.8 ± 2.2 | 31.4 ± 1.4 | 14.7 ± 0.5 | 37.1 ± 1.8 | 33.8 ± 1.9 | 24.4 ± 2.0 | 24.8 ± 1.4 | 25.3 ± 1.2 | 28.2 ± 1.2 | 35.0 |
| Block-10 | 107.2 ± 2.9 | 35.6 ± 1.1 | 18.3 ± 0.9 | 48.3 ± 2.8 | 40.6 ± 2.4 | 56.6 ± 5.5 | 56.7 ± 3.3 | 55.0 ± 3.4 | 61.0 ± 3.1 | 53.3 |
| Block-15 | 111.2 ± 1.9 | **83.9** ± 3.7 | 32.2 ± 2.0 | 48.4 ± 3.0 | 38.4 ± 2.3 | 71.2 ± 5.6 | 70.4 ± 3.5 | 70.0 ± 5.1 | 76.2 ± 3.5 | 66.9 |
| Block-20 | **112.0** ± 2.5 | 78.4 ± 1.9 | **33.4** ± 1.5 | **49.2** ± 2.7 | 41.6 ± 2.6 | **84.7** ± 5.8 | **85.6** ± 2.0 | **84.5** ± 4.2 | **91.7** ± 3.0 | **73.4** |

trend in block size, suggesting that CurrMask progressively enhances masking difficulty. For the mask ratio, CurrMask also continuously adjusts, CurrMask also continuously adjusts, slightly decreasing from 0.55 to 0.54. Through continual adjustments on mask schemes, CurrMask ensures steady improvement on downstream tasks.

### D.2   Impact of Block size

We have conducted experiments on token-wise masking and block-wise masking with constant block sizes of 5,10,15 and 20 to prove the significance of block-wise masking empirically. The results of skill-prompting are in Table D.2.

We can observe from the table that on most tasks, Block-15 and Block-20 achieve higher rewards compared to token-wise masking. Additionally, as the block size increases, the performance of block-wise masking on skill-prompting also improves. These observations support our intuition that block-wise masking with relatively high block sizes helps the model better capture long-term dependencies.

## E   Attention Visualization

In this section, we provide more details about our attention map visualization and additional results.

**Setup**   To provide a clearer visualization of the differences between CurrMask and other baselines in zero-shot skill prompting and goal reaching, we visualize the attention maps of their first layer decoders.For comparison purposes, we employ two masking techniques: prompt masking and goal masking. Prompt masking masks all tokens except the first 8 tokens, while goal masking masks all tokens except two randomly sampled state tokens.

Specifically, we evaluate the aforementioned masking methods on 10 trajectories randomly sampled from validation sets using the pretrained model. We then compute the average attention map for each technique, and finally apply L2 normalization to the attention maps of all four heads to obtain the first layer attention map. We focus on a truncated token sequence of length 32, resulting in a final attention map of size $32 \times 32$ for clearly demonstrating the differences.

**Additional Results**   We provide additional visualization results in Figure 7-8. Apart from the observation that CurrMask better captures long-term dependencies than `Random`, we find that increasing the block size for `Block-wise` leads to greater capabilities in long-term prediction, which supports our intuition regarding the benefits brought by block-wise masking.

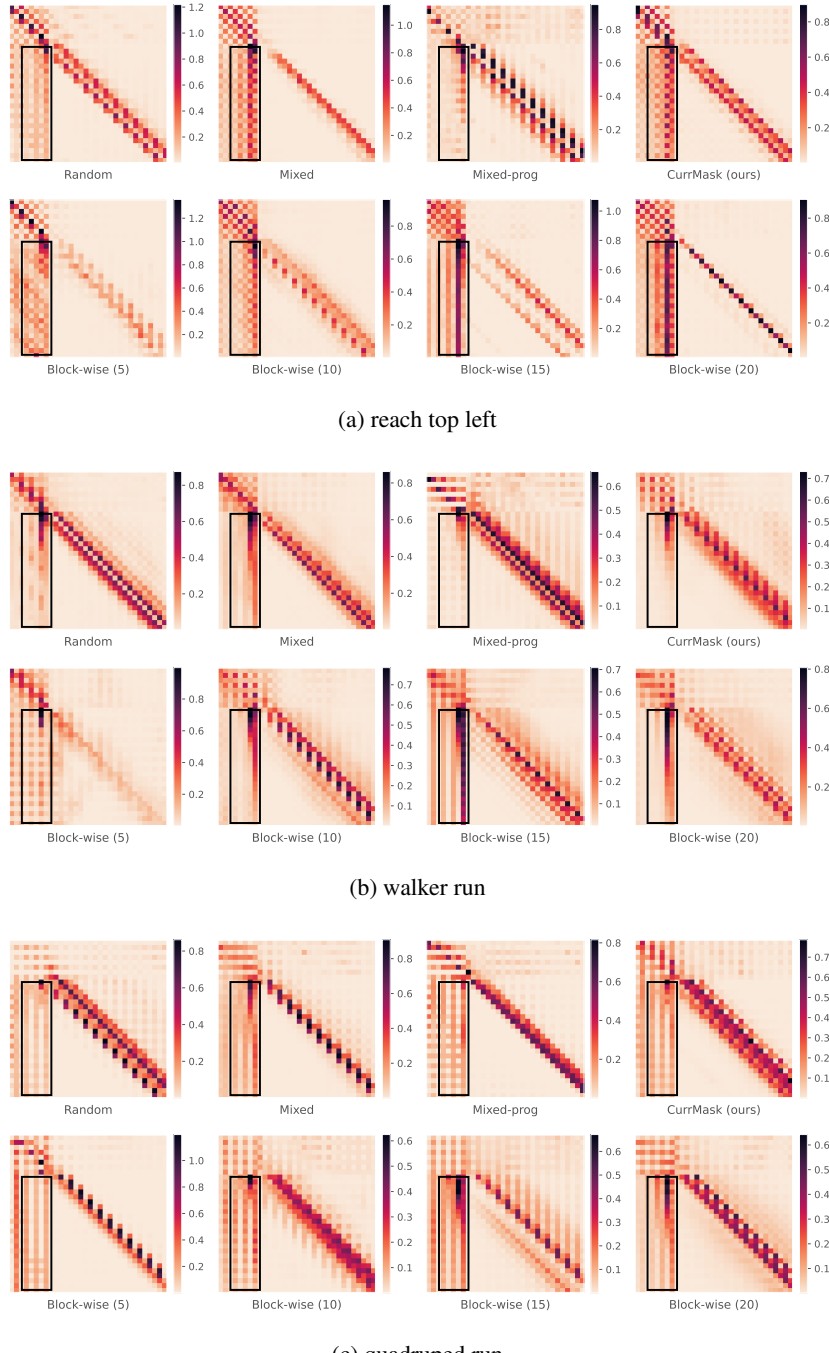

(a) reach top left

(b) walker run

(c) quadruped run

Figure 7: Attention visualization with prompt masking.

## F   Impact Statement

In this paper, we present a curriculum-based masked prediction approach for unsupervised RL pretraining. Although our method is not expected to pose direct social risks, the use of extensive pretraining datasets highlights the significance of avoiding harmful bias in training data. We emphasize the need for ethical responsibility to ensure that our method contributes positively to societal and technological progress.

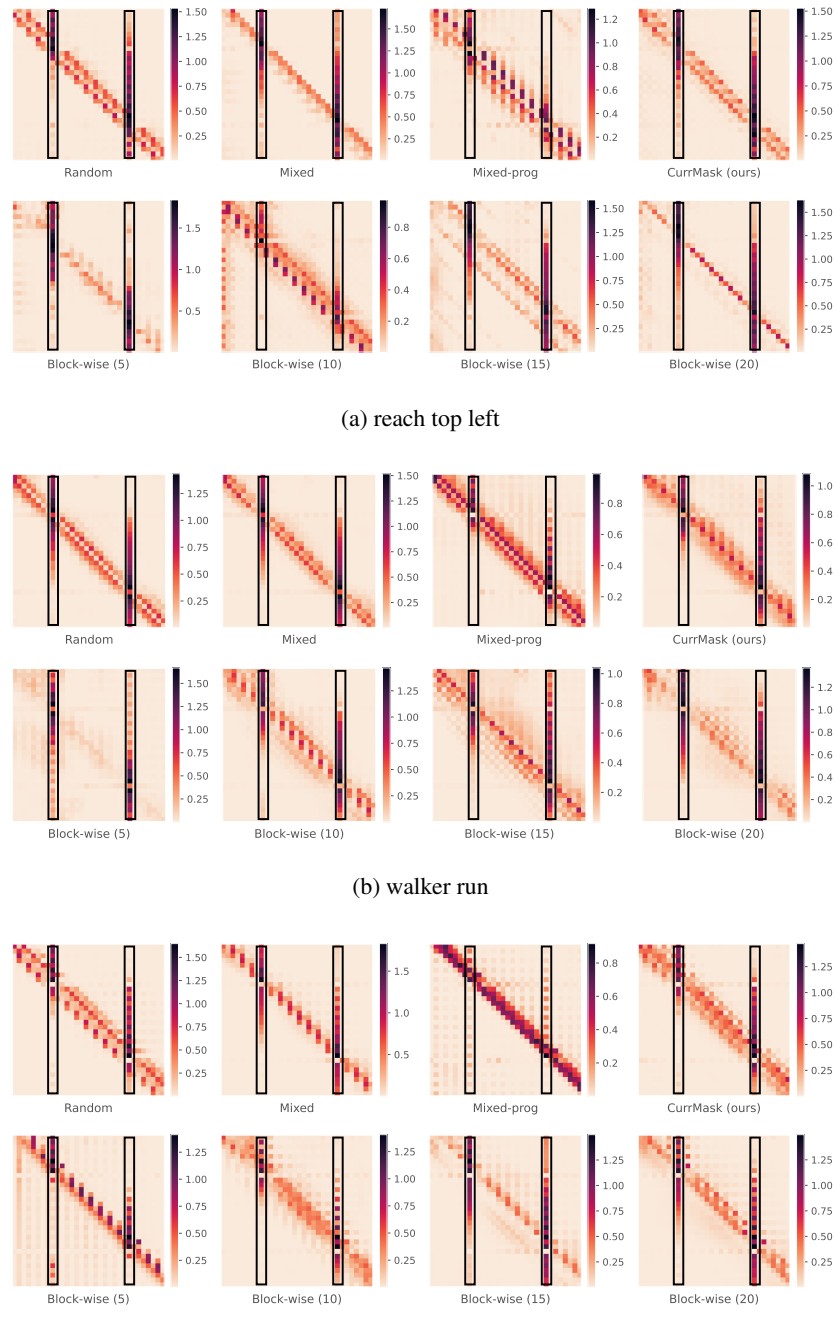

(a) reach top left

(b) walker run

(c) quadruped run

Figure 8: Attention visualization with goal masking.

