# OpenReview forum: "Learning Versatile Skills with Curriculum Masking"
_NeurIPS.cc/2024/Conference — NeurIPS 2024 poster_

### Official Review · Reviewer_Wq3n · 2024-07-07

**Soundness:** 3
**Presentation:** 4
**Contribution:** 3
**Rating:** 6
**Confidence:** 3

**Summary:**

This paper presents CurrMask, a novel masked prediction approach for unsupervised RL pretraining, which learns skills of different complexity through block-wise masking and adaptively adjusts masking schemes in a curriculum for training efficiency. In contrast to previous methods that perform random masking at the token level, CurrMask applies a pool of masking schemes with different block sizes to capture temporal dependencies in various scales. In addition, CurrMask trains a bandit model to schedule the masking schemes to facilitate pre-training using the target loss decrease as the reward. The method is extensively evaluated in three downstream task settings across different environments from the DeepMind control suite. The experimental results demonstrate the strong empirical performance of CurrMask for both zero-shot inference and finetuning, which generally outperforms the compared baselines.

**Strengths:**

- The proposed framework creatively introduces curriculum skill learning to masked prediction for RL pretraining, which facilitates long-term reasoning and training efficiency.
- The proposed method generally brings performance improvements across different downstream tasks.
- Comprehensive analysis is performed to understand the effectiveness of each component in the proposed method.
- The paper is well-written and well-structured.

**Weaknesses:**

- The masked prediction pretraining is interleaved with a bandit training process with non-stationary reward distribution, which could increase the instability of the training process
- A skill usually refers to a meaningful abstraction beyond simply consecutive states and actions (e.g. a state-action sequence completing a subtask/subgoal). Therefore, not all state-action sequences are useful in downstream tasks. Masking blocks randomly may let the model spend lots of capacity on memorizing arbitrary state-action segments, instead of capturing reusable behaviors that can be efficiently transferred to downstream applications.

**Questions:**

- It puzzles me whether block-wise masking is needed necessarily. If mask ratios are randomly chosen from a range, high mask ratios would naturally generate masked blocks of varying sizes, and low mask ratios would also likely ask the model to perform token-level reconstruction, equivalently implementing the mixture of masking schemes, which seems to make the block-wise masking redundant
- It remains unclear to me how Figure 4(a) serves as evidence of long-term prediction capability. And how would the attention map change if the prediction horizon gets longer?
- Given that the reward is computed every $I$ training step(s), how stable the reward calculation and the overall training dynamics is with different $I$s?

**Limitations:**

The authors adequately discussed the limitations of the proposed method.

---

> ### Author Rebuttal · Authors · 2024-08-07
>
> Dear Reviewer Wq3n,
>
> We sincerely appreciate your time to review our paper and your valuable feedback. We will address your concerns in detail below.
>
> >W1 Non-stationary reward distribution may increase instability of the training process
>
> Regarding the concern about non-stationary reward distribution potentially increasing the instability of the training process, we would like to kindly direct you to Appendix C of our paper, where we have a detailed discussion. Despite the non-stationary nature of the reward distribution, our method leverages the inherent adaptability of EXP3 and rescales rewards using historical percentiles to ensure that the learning process progresses gradually without sudden regime shifts.
>
> Additionally, in Appendix D, we have a detailed discussion on the impact of the masking curriculum during the pretraining process. We encourage the reviewer to refer to Figure 5, which demonstrates that CurrMask enables more stable and consistent improvement in skill prompting performance during the pretraining phase compared to other methods.
>
> >W2 Masking blocks let the model spend capacity on memorizing arbitrary state-action segments
>
> We understand your concern that the model may spend capacity on memorizing arbitrary state-action segments. However, during pretraining, it is inherently challenging to determine which information will be effective for downstream tasks, regardless of how we adjust the pretraining paradigm. Given this limitation, our strategy is to explicitly provide a more versatile pool of potential reusable behaviors during pretraining. By doing so, we aim to enhance the model's learning by constructing a masking pool consisting of masking schemes of different mask ratios and block sizes and let the model adaptively adjust the learning progress with curriculum masking.
>
>
> >Q1 Necessity of block-wise masking
>
> To achieve the goal of mixing masking schemes, CurrMask constructs the masking pool with masking schemes that are a combination of high/low ratios and large/small block sizes. Simply mixing different mask ratios intuitively lacks the scenario of combining low mask ratios with large block sizes. To empirically demonstrate the necessity of block-wise masking, we can compare the performance of MaskDP and Mixed on skill-prompting. Both MaskDP and Mixed samples multiple mask ratios. MaskDP leverages token-wise masking and Mixed leverages block-wise masking. Specifically, both MaskDP and Mixed randomly sample a mask ratio from [0.15, 0.35, 0.55, 0.75, 0.95] and Mixed samples a block size from [1,2,...,20] at each pertaining step. The results of skill-prompting are as follows. We **bold** the highest reward in each column.
>
> | Reward $\uparrow$ | walker\_s  | walker\_w | walker\_r  | quad\_w   | quad\_r  | jaco\_bl  | jaco\_br  | jaco\_tl | jaco\_tr  | Average |
> |-------------------|------------------|------------------|------------------|------------------|------------------|------------------|------------------|------------------|------------------|---------|
> | MaskDP| 103.2 ± 2.6 | **58.4 ± 2.3**  | 29.3 ± 1.4  | 36.6 ± 2.2 | 45.1 ± 2.4  | 58.1 ± 4.4 | 58.4 ± 3.0  | 56.9 ± 3.9       | 64.0 ± 3.3    | 56.7    |
> | Mixed | **110.8 ± 2.2** | 54.2 ± 2.0  | **30.5 ± 1.2**   | **43.3 ± 2.7**       | **51.3 ± 2.8**   | **66.0 ± 6.4** | **61.6 ± 3.7**       | **62.3 ± 3.6**       | **66.5 ± 4.0**    | **60.7**    |
>
> Comparing MaskDP and Mixed, we can see that even with mixed mask ratios, applying block-wise masking still benefits the downstream task performance. This observation suggests that merely combining mask ratios cannot replace the effect of leveraging block-wise masking.
>
> >Q2 Explanations of Figure 4(a) and discussion of longer prediction horizon
>
> *Explanations of Figure 4(a)*: We will revise the explanations of Fig. 4a in the main paper. Below are the improved explanations. Fig. 4a shows how CurrMask predicts the attention map for all 32 tokens based on the first 8 unmasked tokens and the subsequent 24 masked tokens. The vertical axis represents the query, and the horizontal axis represents the key. The black-boxed area highlights the model's dependency on the keys of the first 8 unmasked tokens when predicting the attention map for the subsequent 24 masked tokens. It can be observed that the CurrMask's attention in this region is more pronounced, indicating that CurrMask is better at recalling prompt context when predicting the future compared to random masking.
>
> *Longer prediction horizon*: We visualize the attention map where CurrMask predicts the attention map for all 64 tokens based on the first 8 unmasked tokens in Figure 3 of the global rebuttal PDF. The trend that CurrMask's attention in the black-boxed region is more pronounced still holds true.
>
> >Q3 Stability of reward calculation and overall training dynamics with different evaluation interval $I$s
>
> Firstly, in Section 4.3 (L75-179), we introduced our approach of rescaling rewards into the [-1, 1] range using historical percentiles. This ensures that for different $I$s, the reward distribution remains stable within this range.
>
> To assess the impact of different $I$s on overall training dynamics, we choose $I$ from
> [20,50,100(CurrMask),200,500] for visualization. In the global rebuttal PDF, Figure 4 illustrates the pretraining dynamics under these different  $I$s. From the figure, we can observe that the training dynamics are relatively normal for different $I$ values. Comparing the impact of different $I$s on pretraining dynamics, it is evident that larger $I$ values ($I = 200,500$) result in greater fluctuations in loss compared to smaller $I$ values ($I = 20, 50, 100$). This is intuitively understandable, as a larger $I$ means the model pretrains for more steps under the same mask scheme, leading to a greater reduction in loss for the current mask scheme. When the model resamples a new mask scheme after training for $I$ steps, the loss exhibits more noticeable upward fluctuations.

---

> > ### Comment · Reviewer_Wq3n · 2024-08-09
> > **Rebuttal Acknowledged**
> >
> > I'd like to thank the authors for their detailed responses to my concerns and questions. I think a more efficient paradigm for skills pre-training should be explored but I do agree with the author's rationale on this problem. Most of my concerns have been addressed, and I will keep my positive rating.

---

> > > ### Author Response · Authors · 2024-08-10
> > >
> > > Thank you for taking the time to review our work and give insightful advice. We appreciate your suggestion regarding the exploration of more efficient paradigms for skills pre-training and agree that this is a promising direction for future research. We're glad that our responses were able to address most of your concerns, and we are grateful for your positive rating.

---

### Official Review · Reviewer_CzXA · 2024-07-10

**Soundness:** 2
**Presentation:** 2
**Contribution:** 2
**Rating:** 5
**Confidence:** 3

**Summary:**

The paper presents a method that learns skills through curriculum masking. Specifically, the approach CurrMask can automatically arranges the order of different masking schemes for training. The algorithm is tested on Deepmind Control Suite tasks, and show positive results in representation learning, zero-shot skill prompting, and zero-shot goal-conditioned planning.

**Strengths:**

1. The idea of designing different masking curriculum to learn different skills is generally interesting and makes sense.

2. The experiments, although in limited domains do showcase that the method works well for the most part.

3. Overall writing is great. Hyperparameters used in the experiments are provided in the appendix for reproducibility.

**Weaknesses:**

1. My main concern is that there is no comparison to other existing skill learning methods with offline data, e.g. the two papers (Ajay et al., 2021; Jiang et al., 2022) mentioned in the paper's related work section. As a skill learning/discovery paper, at least one of the existing approaches with the similar setting should be empirically compared with.

2. There is not enough explanation for the proposed "task selection" method, which I believe is the central part of the proposed approach. Specifically, in section 4.3, what do \omega_i, \omega'_i, K denote? Does \pi represent the policy? What is the intuition behind the two equations in the task selection subsection? Without explaining these, it is hard for me to understand how the proposed approach learn the masking curriculum.

3. There is no visualization of the proposed masking curriculums. As this is the central contribution of the paper, it would be very interesting to see what the actual masking curriculum is for those continuous control tasks and how is affect the numerical results.

**Questions:**

Figure 1 (a), the inputs are all masked?

**Limitations:**

Yes

---

> ### Author Rebuttal · Authors · 2024-08-07
>
> Dear Reviewer CzXA,
>
> We sincerely appreciate your time to review our paper and your valuable feedback. We will address your concerns in detail below.
>
> >Comparison to existing skill learning methods with offline data
>
> Firstly, we would like to clarify that our paper focuses on masked prediction. Therefore, we aim to compare our method with the most relevant skill learning methods that are also based on masked prediction. In our paper, MaskDP and MTM are the closest baselines that fit this criterion, providing a fair comparison.
>
> Apart from MaskDP and MTM we have provided, regarding the two papers you mentioned, Ajay et al., 2021 and Jiang et al., 2022:
>
> Ajay et al., 2021: We are unable to reproduce the paper, especially without publicly available code.
>
> Jiang et al., 2022: The experimental environments used in this work are grid world and tasks in the visual domain, which are significantly different from our state-based mujoco domain. We follow the hyperparameters in the paper and adapt their method LOVE to our dataset and environment and train 35k steps for a fair comparison. The evaluated results are in the following table.
>
> | Reward $\uparrow$ | walker\_stand | walker\_walk | walker\_run |
> |-------------------|------------------|------------------|------------------|
> | MaskDP |  790.9  ±  141.8   |   439.8   ± 178.9    |   174.7   ±  38.0      |
> | MTM | 786.5  ±   140.3  |   459.8   ± 150.5    |    177.1      ±  39.3  |
> | LOVE | 201.4  ±  122.7   |  135.8   ±  61.7    |  74.6    ±  47.8      |
> | CurrMask |  833.9 ±  89.5   | 699.2   ±   79.4    |   274.4    ±   61.3    |
>
> From the table, we can observe that LOVE cannot perform reasonable results under our experimental setting and requires further tuning. We hope this clarifies our rationale and the efforts we made to provide a fair and relevant comparison in our paper. To enhance clarity, we will revise our paper to include more detailed considerations regarding the selection of baselines.
>
> >Explanations on notations and intuition of the "task selection" method
>
> *Notations in Section 4.3*:
>
> $\omega_i$ and  $\omega'_i$: $\omega_i$ represents the weight of the $i$-th arm (or masking scheme) and $\omega'_i$ represents the updated $\omega_i$  in Equation 4.
>
> $K$: This denotes the number of arms (or masking schemes) available in the masking scheme pool $\mathcal{M}$.
>
> $\pi$: This represents the sampling probability distribution for each masking scheme.
>
> *Intuition of Equation 3 and 4*:
>
> Equation 3: This equation shows how to sample each arm (i.e., each masking scheme) using weights. The weights $\omega$​ determine the probability distribution $\pi$. We sample the masking schemes based on $\pi$. This probabilistic approach allows us to balance exploration and exploitation by dynamically adjusting the focus on different masking schemes based on their performance.
>
> Equation 4: This equation describes how we update the arm weights based on the observed rewards. By updating the weights $\omega$ according to the rewards, we ensure that the more effective masking schemes (those that lead to better pretraining progress) are sampled more frequently in subsequent iterations. This adaptive mechanism helps the model to learn an effective masking curriculum over time.
>
> Thanks again for providing the detailed suggestions, and we will rewrite this section in the updated version accordingly.
>
> > Visualization of masking curriculum
>
> We would like to clarify that we have included the visualization curriculum masking and the analysis of the impact on performance in Appendix D. We kindly ask the reviewer to refer to the section for more details. We also include a more detailed visualization of the masking curriculum in Figure 1 of the global rebuttal pdf.
>
> From Appendix D and Fig. 1 in the global rebuttal pdf, we can see that during the pretraining process, CurrMask continually adjusts the block size and mask ratio according to the training progress. Overall, there is an upward trend in block size, suggesting that CurrMask progressively enhances masking difficulty. For the mask ratio, CurrMask also continuously adjusts, CurrMask also continuously adjusts, slightly decreasing from 0.55 to 0.54. Through continual adjustments on mask schemes, CurrMask ensures steady improvement on downstream tasks. We will revise the paper to highlight this aspect thanks to this advice.

---

> > ### Comment · Reviewer_CzXA · 2024-08-10
> >
> > Thank you for your detailed response. My concerns have been addressed and I will keep my overall positive score.

---

> > > ### Author Response · Authors · 2024-08-11
> > >
> > > Thank you very much for your positive score. We are happy to discuss further if there are any additional questions.

---

### Official Review · Reviewer_znVZ · 2024-07-11

**Soundness:** 3
**Presentation:** 3
**Contribution:** 2
**Rating:** 5
**Confidence:** 3

**Summary:**

This work proposes a curriculum learning approach to reinforcement learning skills from masked trajectory sequences. Given a set of pre-collected environment samples (here from a TD3 agent in 9 different mujoco domains), the proposed MaskCurr curriculum treats the agent's progress (target-loss decrease) like a two-armed bandit problem, where one of the bandits is the amount of information that is covered up (mask-ratio) and the other bandit is the length of the covered gaps that the agent has to reconstruct (block-size). Evaluation is tested on two tasks, skill-prompting, which requires the trained agent to complete a starting sequence, and goal-planning, where masked trajectories with interspersed checkpoints (goals) have to be filled in such that the intermediate goals are reached. CurrMask is compared to a set of various masking techniques (i.e., variations of random masking) as well as GPT variants, and is shown to perform competitively in both of the above mentioned tasks.

**Strengths:**

- Well written paper that is easy to follow, with clean formalization and good, readable balance between text and material (plots, diagrams, tables, algorithms).
- Generally decent evaluation, 9 environments / tasks, each evaluated with 20 runs. Fine-tuning potential included as well, although not quite fairly assessed (see weaknesses).
- The choice of random-masking baseline variants does cover a good range of the ablation information of CurrMask, and the results analysis (Tab.1, Fig.3&4) provides insightful understanding of the experiments.

**Weaknesses:**

- Not clear how "versatile" or "diverse" skills are classified here. I understand these properties w.r.t. to how different single tasks behaviors are emerging, rather than "learn multiple tasks", which I believe is meant here.
- The fine-tuning experiments (Fig.2) seem to be plotted unfairly against the "from scatch" baseline, since pretrainined models do not "start" at step 0 (but at -#pretraining-steps) on the x-axis. For the training itself, 25k steps for mujoco domains is rather brief and all training curves seem to be stopped mid-training. As such the actual suitability for fine-tuning is rather questionable, or at least not shown with significance.
- The attention-map result of Fig.4 a) could be better interpreted in the main paper, there is a visual difference but its not quite clear to me how these maps correlate to the claimed useful long-term dependencies skills. The appendix provides some more insight, but all relevant information and explanations of main paper plots should be in the main paper as well.
- Experiments could use some different domain for comparison (apart from the mujoco domains).
- While the improvement of CurrMask compared to the baselines is shown, the significance of the results is lowered by the fact on how good the random masking techniques still perform.


---
Minor issues:
- Repeated mentions to "following previous/prior work Liu et al.2022" which give the impression of self-reference. If this is not a self-citation please clarify the wording for the double blind reviewing standard.
- l26 missing word "conditioned on the remaining (?), ..."
- The title could be more specific, claiming generally versatile skills for (basically only) the mujoco domain feels a bit far-fetched. The scope of this work is not broad enough to warrant such a sweeping claim.

**Questions:**

- The "from scratch" baseline is TD3?
- Although CurrMask mostly outperforms the random baselines, did you perhaps try factor in the overhead of the CurrMask learning into the evaluation? I.e., if the random variants would increase their training time by the observed 4.7% wall-clock time overhead, how would your estimation be on the performance comparison given in Table 1?

---
Edit after rebuttal: Questions have been addressed, I will keep the overall positive score.

**Limitations:**

Apart from the mentioned overhead the proposed method is rather straightforward and simple, i.e., not many limitations.

---

> ### Author Rebuttal · Authors · 2024-08-07
>
> Dear Reviewer znVZ,
>
> Thank you for your insightful review of our work! We appreciate your valuable suggestions and will respond to your questions in detail.
>
> >W1 Explanations on "versatile" or "diverse" skills
>
> In our study, we use the terms "versatile skills" or "diverse skills" to refer to the model's ability to learn multiple skills. According to the dictionary definition, versatile means "able to adapt or be adapted to many different functions or activities". In our context, versatile skills indicate that through CurrMask, the pretrained model becomes adaptable to different potential downstream tasks. We will clarify this in our updated paper.
>
> > W2 Clarifications on the starting step and number of training steps in fine-tuning experiments (Fig.2)
>
> *The starting step in Fig.2*: In the fine-tuning experiments, we compare the pretrained baselines with the “from scratch” baseline to see whether the pretraining process is helpful to offline RL. Therefore, all the baselines should be compared using the same fine-tuning steps (instead of pretraining steps) as the x-axis, starting from 0. Such comparison is common in previous offline RL pretraining works[1][2].
>
> *Training steps*: Our model architecture, data collection, pretraining, and finetuning protocols follow the previously established protocols in previous work[1], where the number of finetuning steps was set to 25k and also experimented on similar mujoco datasets.
>
> >W3 Interpretation of Fig.4a
>
> Thanks for pointing this out. We will revise the explanations of Fig. 4a in the main paper. Below are the improved explanations:
> "Fig. 4a shows how CurrMask predicts the attention map for all 32 tokens based on the first 8 unmasked tokens and the subsequent 24 masked tokens. The vertical axis represents the query, and the horizontal axis represents the key. The black-boxed area highlights the model's dependency on the keys of the first 8 unmasked tokens when predicting the attention map for the subsequent 24 masked tokens. It can be observed that the attention of CurrMask in this region is more pronounced, indicating that CurrMask is better at recalling prompt context when predicting the future compared to random masking."
>
> >W4 Experiments on domains apart from mujoco
>
> Thank you for your suggestion regarding experiments on domains apart from Mujoco. While our current experiments focus on the Mujoco domain, we will consider extending our evaluation to other domains in future work.
>
> >W5 Performance of random masking reduces the significance of CurrMask
>
> We would like to highlight that the effectiveness of CurrMask is demonstrated by its consistent outperformance across various tasks, indicating its ability to better capture long-term dependencies and improve model generalization. This advantage is particularly evident in scenarios where capturing complex patterns, such as long-distance predictions, is crucial, proving CurrMask to be a highly valuable technique.
>
> >Minor issues
>
> We appreciate your attention to detail and pointing out all the minor issues.
>
> *Repeated mentions of previous work*: We would like to clarify that our primary goal in mentioning the work is to highlight the established protocols and settings we followed to ensure a fair comparison. Thanks to your advice, we will revise the wording to avoid repeated mentions in the paper.
>
> *l26 Missing Word*: Thanks for pointing out the missing word. We will rewrite this sentence for clarity as follows: “By masking a portion of the input trajectory and predicting conditioned on the remaining unmasked tokens, …”.
>
> *Title Specificity*: We appreciate your suggestion regarding the title. To better reflect the scope of our work, we will consider revising the title to be more specific to our focus on RL pertaining.
>
> >Q1 The "from scratch" baseline is TD3?
>
> Yes.
>
> >Q2 Factor in CurrMask learning overhead
>
> It is indeed a very interesting aspect! We experiment on MaskDP and Mixed  see the impact of the overhead computation, as the training time for CurrMask is 5% higher than these baselines. Specifically, we assess how the performance of these baselines would change if their training steps were increased by 5% (from 300k pretraining steps to 315k). The results are summarized as follows:
>
> | Reward $\uparrow$ | walker\_s  | walker\_w| walker\_r  | quad\_w  | quad\_r   | jaco\_bl  | jaco\_br  | jaco\_tl | jaco\_tr  | Average |
> |-------------------|------------------|------------------|------------------|------------------|------------------|------------------|------------------|------------------|------------------|---------|
> |MaskDP_300k   | 103.2 ± 2.6      | 58.4 ± 2.3       | 29.3 ± 1.4       | 36.6 ± 2.2       | 45.1 ± 2.4     | 58.1 ± 4.4       | 58.4 ± 3.0       | 56.9 ± 3.9       | 64.0 ± 3.3       | 56.7    |
> |MaskDP_315k  |  103.1 ± 2.7 |52.4 ± 2.3 | 30.6 ± 0.9 | 39.3 ± 2.7 | 47.0 ± 2.8   |  59.2 ± 4.6             | 59.6 ± 3.3              | 59.5 ± 5.2          | 68.3 ± 3.7  | 57.6    |
> |Mixed_300k| 110.8 ± 2.2 | 54.2 ± 2.0 | 30.5 ± 1.2 | 43.3 ± 2.7 | 51.3 ± 2.8 | 66.0 ± 6.4 | 61.6 ± 3.7 | 62.3 ± 3.6 | 66.5 ± 4.0 | 60.7 |
> |Mixed_315k| 109.7 ± 2.3 | 55.4 ± 2.1 | 32.1 ± 1.7 | 43.1 ± 2.4 | 53.2 ± 3.1 | 67.2 ± 5.7 | 63.0 ± 4.2 | 65.3 ± 3.3 | 67.7 ± 4.1 | 61.9 |
> |CurrMask_300k    | 111.2 ± 2.4 | 79.9 ± 1.2 | 38.9 ± 1.9 | 38.0 ± 2.2 | 51.0 ± 3.4 | 88.4 ± 5.1 | 88.5 ± 3.6 | 86.0 ± 4.3 | 92.9 ± 3.5 | 75.0 |
>
> We can see that from 300k to 315k pretraining steps, the average reward improvement of MaskDP and Mixed is very limited (within 2%). After accounting for the overhead computation factor, their performance still lags behind CurrMask, demonstrating that CurrMask's benefits are not solely due to additional computation time.
>
> [1]Liu F, Liu H, Grover A, et al. Masked autoencoding for scalable and generalizable decision making.
>
> [2]Wu P, Majumdar A, Stone K, et al. Masked Trajectory Models for Prediction, Representation, and Control

---

> > ### Comment · Reviewer_znVZ · 2024-08-11
> > **Rebuttal Acknowledgement**
> >
> > I'd like to thank the authors for taking the time and discussing the weaknesses in detail. I think the overall positive trend leaning accept is adequate and I will keep my score.

---

> > > ### Author Response · Authors · 2024-08-11
> > >
> > > We would like to thank you for your positive response. Please let us know if we have addressed all your concerns or if you have any further questions, and we will respond promptly.

---

### Official Review · Reviewer_f2Vb · 2024-07-15

**Soundness:** 4
**Presentation:** 3
**Contribution:** 3
**Rating:** 5
**Confidence:** 4

**Summary:**

This paper proposes a curriculum masking pretraining paradigm for RL training, which is based on the block-wise masking schemes and is able to decide the block size and mask ratio automatically. Specifically, the authors design a masking pool with different masking scheme of different block size and mask ratio. Given the target loss and the corresponding reward from the environment, this method formulates the masking selection as a multi-armed bandit problem and sample the masking scheme from the masking pool according to the updated policy. The experiments on control tasks demonstrate the effectiveness of this method.

**Strengths:**

1.	The paper is well written and easy to follow.
2.	This work uncovers that the optimal combination of block size and mask ratio requires adaptive selection.
3.	The analytical experiments demonstrate the ability of this method to capture long-term dependencies.

**Weaknesses:**

1.	The authors have not conducted experiments to compare token-wise and block-wise masking before they choose the block-wise masking. Despite the theoretical analysis in the Introduction section, it would be better to prove this choice through experimentation.
2.	There is no time complexity analysis of this method. If possible, the authors could present how much time each baseline and this method consume respectively.
3.	In figure 6, the authors report the mean block size of the whole training process, so I am curious about the mean mask ratio of the whole training process. Furthermore, is it possible to visualize the impact of mask ratio just like Figure 3(a), so that we can know whether masked prediction benefits from larger mask ratio or smaller ratio.

**Questions:**

See weaknesses.

**Limitations:**

The limitations are discussed in the last section of this paper. There is no potential negative societal impact.

---

> ### Author Rebuttal · Authors · 2024-08-07
>
> Dear Reviewer f2Vb,
>
> Thank you for your valuable feedback! We appreciate your positive remarks about the soundness, presentation, and contribution of our work. Below, we address the specific points you raised:
> > Comparing token-wise and block-wise masking
>
> We have conducted experiments on token-wise masking and block-wise masking with constant block sizes of 5,10,15 and 20 to prove the significance of block-wise masking empirically. We will include this experiment in our revised appendix. The results of skill-prompting are in the following table. We **bold** the highest value and *italicized* the second highest value in each column.
>
> | Reward $\uparrow$ | walker\_s        | walker\_w        | walker\_r        | quad\_w          | quad\_r          | jaco\_bl         | jaco\_br         | jaco\_tl         | jaco\_tr         | Average |
> |-------------------|------------------|------------------|------------------|------------------|------------------|------------------|------------------|------------------|------------------|---------|
> | Token             | 103.2 ± 2.6      | 58.4 ± 2.3       | 29.3 ± 1.4       | 36.6 ± 2.2       | **45.1 ± 2.4**     | 58.1 ± 4.4       | 58.4 ± 3.0       | 56.9 ± 3.9       | 64.0 ± 3.3       | 56.7    |
> | Block-5           | 95.8 ± 2.2       | 31.4 ± 1.4       | 14.7 ± 0.5       | 37.1 ± 1.8       | 33.8 ± 1.9       | 24.4 ± 2.0       | 24.8 ± 1.4       | 25.3 ± 1.2       | 28.2 ± 1.2       | 35.0    |
> | Block-10          | 107.2 ± 2.9      | 35.6 ± 1.1       | 18.3 ± 0.9       | 48.3 ± 2.8   | 40.6 ± 2.4       |  56.6 ± 5.5       | 56.7 ± 3.3       | 55.0 ± 3.4       | 61.0 ± 3.1   |  53.3   |
> | Block-15          | *111.2 ± 1.9*    | **83.9 ± 3.7**   | *32.2 ± 2.0*     | *48.4 ± 3.0*     | 38.4 ± 2.3       | *71.2 ± 5.6*     | *70.4 ± 3.5*     | *70.0 ± 5.1*     | *76.2 ± 3.5*     |  *66.9*   |
> | Block-20          | **112.0 ± 2.5**  | *78.4 ± 1.9*     | **33.4 ± 1.5**   | **49.2 ± 2.7**   | *41.6 ± 2.6*     | **84.7 ± 5.8**   | **85.6 ± 2.0**   | **84.5 ± 4.2**   | **91.7 ± 3.0**   | **73.4**|
>
> We can observe from the table that on most tasks, Block-15 and Block-20 achieve higher rewards compared to token-wise masking. Additionally, as the block size increases, the performance of block-wise masking on skill-prompting also improves. These observations support our intuition that block-wise masking with relatively high block sizes helps the model better capture long-term dependencies.
>
> >  Time complexity analysis
>
> Firstly, we would like to clarify that we briefly compare the training time overhead of CurrMask versus random masking (MaskDP) in the last line of page 9. Specifically, we observed a training time (wall clock time) overhead of 4.7% for 100k gradient steps.
> We wholeheartedly agree with the reviewer's suggestion to present the exact time consumption for each baseline and our method. To address this, we have run all the baselines on a single NVIDIA 3090 GPU and recorded pertaining time for 300k steps, as shown in the table below.
>
> | Baseline     | MaskDP |   MTM   |   Mixed   | Mixed-prog | Mixed-inv | CurrMask |
> |------------|-----------|-----------|-----------|-----------|-----------|-----------|
> | Time (seconds) | 27,450  | 28,457| 27,733 | 27,864     | 27,931    | 28,998    |
>
> All the baselines in the table share the same model architecture. From this table, we can see that the pretraining time of CurrMask is approximately 5% longer than the fastest baseline, MaskDP. This additional time is mainly due to the overhead of evaluating pretraining progress every 100 steps and adjusting the mask scheme sampling distribution accordingly.
>
> > Visualizations of mean mask ratio in the training process and impact of mask ratio
>
> For the added figure, we submit a PDF file in the global rebuttal. We kindly ask you to refer to the PDF for detailed figures.
>
> We have included both the mean mask ratio and mean block size of the whole training process in **Fig. 1**. We can see that during the pretraining process, CurrMask continually adjusts the block size and mask ratio according to the training progress. Overall, there is an upward trend in block size, suggesting that CurrMask progressively enhances masking difficulty to better capture the long-term dependency. For the mask ratio, CurrMask also continuously adjusts with the mean mask ratio slightly decreasing from 0.55 to 0.54.
>
> We also include the visualization of the impact of mask ratio on mask prediction in **Fig. 2**. Similar to Figure 3(a) in the main paper, we compare masking with constant mask ratios 0.15,0.35,0.55,0.75,0.95  with masking with Mixed_ratios and CurrMask. Mixed_ratio uniformly samples a mask ratio from [0.15,0.35,0.55,0.75,0.95] in each step. We can observe that when sampling with a constant mask ratio, a larger mask ratio tends to benefit masked prediction more. However, when sampling with mixed mask ratios, the performance surpasses all baselines that sample with a constant mask ratio. This indicates that low and high mask ratios have irreplaceable benefits for masked prediction, making sampling from multiple mask ratios an appropriate choice. This observation is similar to the conclusion  mentioned in previous work[1] that the mixed mask ratio strategy outperforms fixed mask ratio baselines.
>
> [1]Liu F, Liu H, Grover A, et al. Masked autoencoding for scalable and generalizable decision making.

---

### Author Rebuttal · Authors · 2024-08-07

We give our sincere thanks to all the reviewers for their insightful suggestions and positive feedback of our work. For all the figures that we add for further discussion, we submit a PDF file in this global rebuttal.

---

### Decision · Program_Chairs · 2024-09-25

**Decision:**

Accept (poster)

**Comment:**

The paper presents a curriculum learning method for RL, CurrMask, that learns skills from masked trajectories. The reviewers found the paper, among others,  well written, the idea appealing, and performance improvements on the downstream tasks. After the rebuttal, three out of four reviewers found most of their concerns addressed (the remaining reviewer did not respond), with all ratings keeping a positive trend leaning toward acceptance. Therefore, I recommend the paper to be accepted.